# Emergence of directional bias in tau deposition from axonal transport dynamics

**Justin Torok** [1] *, **Pedro D. Maia** [2], **Parul Verma** [3], **Christopher Mezias** [3], **Ashish Raj** [3] *

**1** Department of Computational Biology and Medicine, Weill Cornell Medical School, New York, New York, United States of America, **2** Department of Mathematics, University of Texas at Arlington, Arlington, Texas, United States of America, **3** Department of Radiology and Biomedical Imaging, University of California at San Francisco, San Francisco, California, United States of America

* jut2008@med.cornell.edu, jlt46@cornell.edu (JT); ashish.raj@ucsf.edu (AR)

## Abstract

Defects in axonal transport may partly underpin the differences between the observed pathophysiology of Alzheimer's disease (AD) and that of other non-amyloidogenic tauopathies. Particularly, pathological tau variants may have molecular properties that dysregulate motor proteins responsible for the anterograde-directed transport of tau in a disease-specific fashion. Here we develop the first computational model of tau-modified axonal transport that produces directional biases in the spread of tau pathology. We simulated the spatiotemporal profiles of soluble and insoluble tau species in a multicompartment, two-neuron system using biologically plausible parameters and time scales. Changes in the balance of tau transport feedback parameters can elicit anterograde and retrograde biases in the distributions of soluble and insoluble tau between compartments in the system. Aggregation and fragmentation parameters can also perturb this balance, suggesting a complex interplay between these distinct molecular processes. Critically, we show that the model faithfully recreates the characteristic network spread biases in both AD-like and non-AD-like mouse tauopathy models. Tau transport feedback may therefore help link microscopic differences in tau conformational states and the resulting variety in clinical presentations.

## Author Summary

The misfolding and spread of the axonal protein tau is a hallmark of the pathology of many neurodegenerative disorders, including Alzheimer's disease and frontotemporal lobar dementia. How tau misfolding causes disorders with distinct neuropathology and clinical presentations is the subject of ongoing research. Although current evidence suggests that the specific conformations tau adopts affect where it spreads throughout the brain, a mechanistic explanation has remained elusive. Here we propose that the conformer-specific dysregulation of axonal transport can lead to directionally biased spread, and we employ a mathematical model to explore how tau spreads between neurons in the context of this transport feedback mechanism. We find that conformation-specific feedback is flexible enough to explain bias in either direction and thoroughly explore how this bias emerges as a function of the model's key parameters. Further, the model

**Data Availability Statement:** All code for running the model and the analysis pipeline are contained in the following GitHub repository: https://github.com/Raj-Lab-UCSF/Tau_Transport/. All data files are hosted on the author's cloud for public use:

https://drive.google.com/file/d/1ZYeZ99V5h8n2cUvYBUiF2TOeKVAtHc65/view.

**Funding:** This work was funded through Grant #RF1AG062196 from the NIH National Institute on Aging (NIH-NIA) awarded to AR. The funders had no role in study design, data collection and analysis, decision to publish, or preparation of the manuscript.

**Competing interests:** The authors have declared that no competing interests exist.

reproduces the temporal evolution of directionality observed in two classes of *in vivo* tauopathy models, demonstrating that transport feedback is sufficient to explain differential tau spread as a function of conformation.

## 1 Introduction

Despite manifest differences in clinical presentation, the pathology of many progressive neurodegenerative diseases is linked to aberrant protein misfolding and aggregation. Indeed, in Alzheimer's disease (AD) [1], frontotemporal lobar degeneration (FTLD) [2], Parkinson's disease [3], Huntington disease [4], and in amyotrophic lateral sclerosis [5], there are one or more proteinaceous species that appear to spread throughout the brain and induce neuronal dysfunction and death. Both *in vitro* experiments with cultured neurons [6–10] and *in vivo* work with animal models of neurodegeneration [11–13] have confirmed that these prion-like assemblies can spread directly between neurons by traveling within axonal fibers and traversing the synapse. Although the mechanisms by which these misfolded protein agents arise, propagate, and contribute to pathology remain incompletely understood, they do play a central role in the progression of neurodegenerative diseases [14, 15].

Tauopathy, the pathological accumulation and spread of misfolded aggregates of microtubule-associated protein tau, is among the most common types of proteinopathies implicated in neurodegenerative disease, including AD, FTLD, Pick's disease, and others. The recent discovery that there are phosphoepitopes of tau that distinguish AD from FTLD-type disorders suggests a direct link between the specific *conformer* of misfolded tau and disease diagnosis [16–18]. Heterogeneity in tau hyperphosphorylation and misfolding also exists between AD patients, giving rise to distinct clinical phenotypes [19]. An appealing hypothesis for explaining how specific tau conformations lead to distinct patterns of neurodegeneration is that although these tau species can all spread between connected brain regions via white matter tracts, their propensities to travel in the anterograde (i.e. from presynaptic to postsynaptic cell) or retrograde (i.e. postsynaptic to presynaptic) directions may not be equal. Recent work by our group has lent further support for this hypothesis, wherein we found that mouse tauopathy models with tau species that developed in the presence of amyloid-β showed a marked preference to migrate in the retrograde direction, while no consensus directional bias emerged among amyloid-negative models [20]. However, there are no models to date that explore *how* this directional bias arises.

Mechanistically, conformer-specific dysregulation of active transport could explain differential biases in the direction of tau spread between diseases. In healthy neurons, energy-driven active transport within the axon is governed by the complex interplay of motor proteins, their cargoes, and the microtubules along which they travel [21, 22]. Early in AD, hyperphosphorylated tau misfolds and aggregates in the axon; as the disease progresses, it causes the breakdown of the axon initial segment barrier and enters the somatodendritic compartment [23–25]. *In vitro* experiments provide direct evidence that active transport is regulated by axonal tau concentration: a primary anterograde-directed motor protein, kinesin-1, is physically obstructed by microtubule-bound tau, while the retrograde-directed motor protein, dynein, remains mostly unaffected except at aphysiological concentrations [26, 27]. Disease-mimicking tau variants limit this endogenous ability to inhibit kinesin-1 [28–30], while neurofibrillary tangles (NFTs) knock down kinesin-1 [31]. Since anterograde-directed and retrograde-directed motor proteins are in a constant state of "tug-of-war" whose outcome ultimately governs the direction and rate of cargo transport, including that of tau itself [27, 29, 32], the

pathological effects of tau on kinesin-1 specifically provide an underexplored basis for the conformer-specific spatial patterns of NFTs observed *in vivo* and clinically. Using a computational model to simulate the effects of kinesin regulation by tau on its spatial deposition patterns, therefore, could provide the needed conceptual link between the biochemical properties of specific tau conformers and the observed directional spread bias. While detailed models of tau axonal transport exist [33], there have been no explorations to date of the consequences of pathological tau feedback on axonal transport.

In the present work, we model two distinct species of pathological tau in a closed two-neuron system under biologically plausible conditions, demonstrating the impact of transport feedback has on their spatial concentration profiles over time. **This model yields the following insights: (i)** The balance of transport feedback parameters alone is sufficient to develop strong directional biases in both the anterograde and retrograde directions. **(ii)** The rates of aggregation and fragmentation in the model exert a dramatic effect on the steady-state configuration of the system as well, indicating that processes of interconversion between soluble and insoluble tau species are inextricably linked to how transport feedback influences the system. **(iii)** This nonlinear system converges to a steady state over a wide range of initial conditions. **(iv)** Despite its simplicity, the transport feedback model produces biases that quantitatively match those that develop at the network level both in AD-like and non-AD-like mouse tauopathy models [20]. Taken together, these results demonstrate that a simple transport feedback mechanism can explain how different tau conformers, which have unique molecular properties, can develop distinct directional biases and propagate differentially across the brain.

## 2 Results

### 2.1 Model scope

We construct a system of partial differential equations (PDEs) to model the concentration profiles along a single spatial axis of two biophysically distinct species of pathological tau: *soluble tau*, such as misfolded monomers and small oligomers, and *insoluble tau*, such as larger filaments and tangles. See Fig 1 for a full schematic of the system. The dynamics of each species depend upon the biological compartment in which it resides: the presynaptic and postsynaptic somatodendritic (SD) compartments, the axon, the axon initial segment (AIS), and the synaptic cleft (SC) are modeled distinctly. These two tau species can interconvert through the opposing processes of fragmentation and aggregation (Eq 5). For both biophysical and mathematical (see for instance the work of Kuznetsov and Kuznetsov [35]) reasons, we model the effective transport of the mobile, soluble species as two component processes: diffusion and active transport, the latter of which only occurs within the axonal compartment. The effective velocity of active transport within the axon, $v$ (Eq 1), is a function of the local concentrations of the soluble and insoluble species, because each can perturb the effective anterograde transport rate of kinesin [28, 29, 31]. Specifically, we allow soluble tau to increase effective anterograde transport velocity through the parameter $\delta$ and insoluble tau to decrease it through the parameter $\epsilon$ (Table 1). Since the effective retrograde transport velocity is unaffected by pathological tau [26, 27], directional bias of tau transport is governed by the balance of $\delta$ and $\epsilon$ acting upon anterograde transport rate. Diffusion between the SD and axonal compartments is hindered by the physical barriers of the AIS and SC, leading to slow, concentration-gradient-dependent mixing. We thoroughly explore the dependence of model behavior on these two transport-specific parameters below; see Methods: **Model Description** for a complete explanation of the development and implementation of our model and Table 2 for a summary of the processes modeled in each biological compartment.

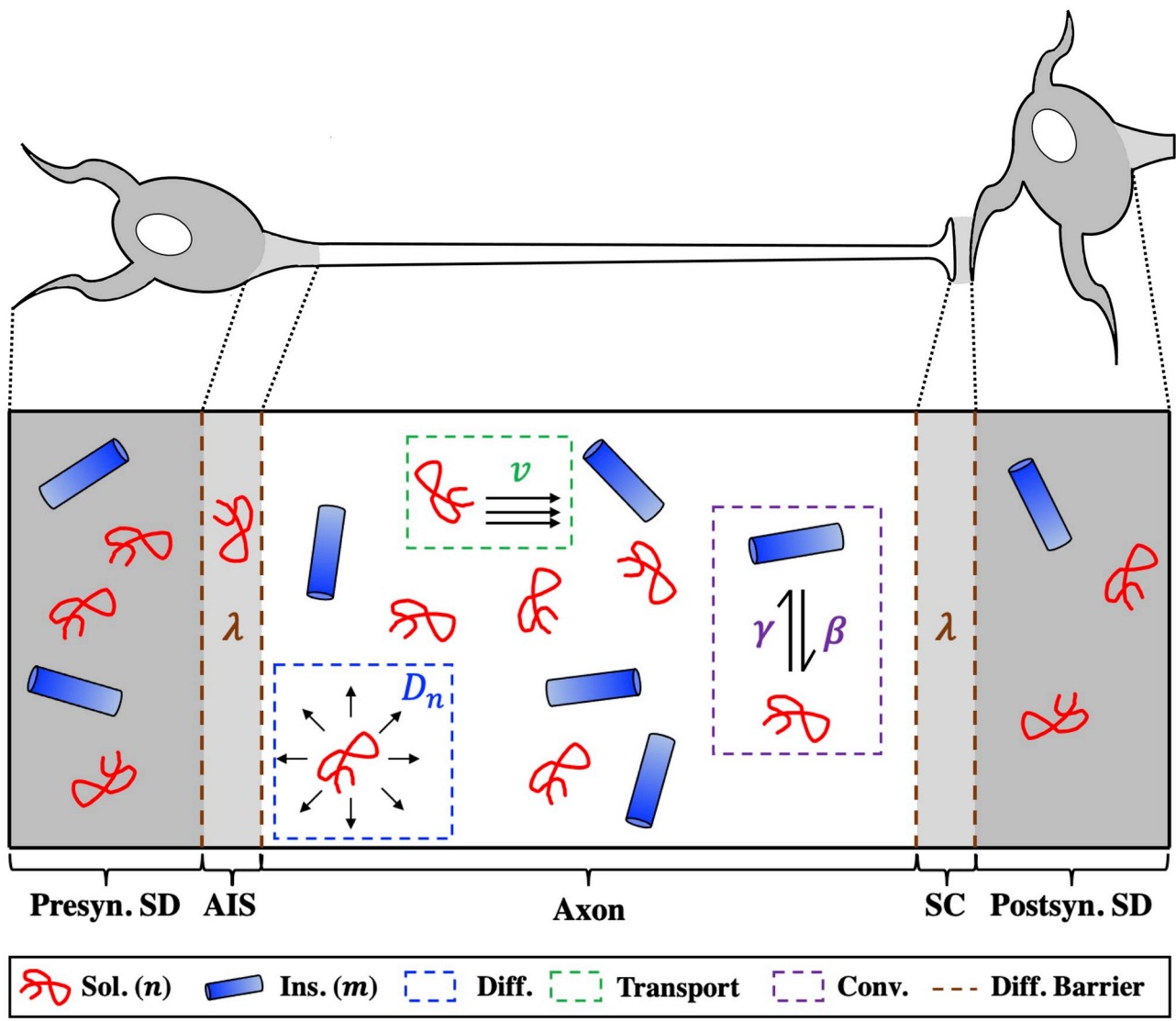

**Fig 1. Model System.** Schematized version of the one-dimensional system that we simulate. We model two distinct species of pathological tau, soluble (red) and insoluble (blue), across within a multi-compartment model mimicking the two-neuron system shown in the top panel. The main biological phenomena captured in this model are diffusion (blue box), active transport (green box), species interconversion through fragmentation and aggregation (purple box), and a diffusion-based barrier to inter-compartmental spread (brown dashed lines).

## 2.2 Model regimes

We first explored how the balance of $\delta$ and $\epsilon$ affected the dynamics of tau in three distinct regimes. For each instance, we initiated the model with a uniform concentration of soluble tau $n$ in the axonal compartment only and zero insoluble tau $m$ anywhere. See S1 Table for a full list of the parameters used for each condition.

**2.2.1 Anterograde-biased regime.** Fig 2a and S1 Video show the simulation results for our two-species model with $\delta \gg \epsilon$, which emulates a condition where soluble tau exerts a strong effect on enhancing kinesin processivity but insoluble tau has a minimal effect on inhibiting it. We depict the concentrations of $n$ and $m$, as well as the calculated concentration flux of $n$, $j_{net}$ (obtained by numerically evaluating Eq 4 post hoc), at five time points between model

**Table 1. Glossary of symbols used throughout the text.** Values marked with an asterisk were estimated by Konzack *et al.* [34] in cultured rodent neurons. Ant. = anterograde, ret. = retrograde, conc. = concentration, vel. = velocity.

| Symbol | Description | Remark |
|---|---|---|
| $x$ | Space | $x \in [0, L_{\text{total}}]$, where $L_{\text{total}}$ is the total size of the system in $\mu m$ |
| $t$ | Time | $t \in [0, T]$, where $T$ is the time to reach steady state in s |
| $n(x, t)$ | Soluble tau (conc.) | Number of monomeric units per volume within the fiber in $\mu M$ |
| $m(x, t)$ | Insoluble tau (conc.) | Number of monomeric units per volume within the fiber in $\mu M$ |
| $D_n$ | Theoretical diffusivity of $n$ | Estimated to be 12 $\mu m^2/s^*$ |
| $f$ | Diffusing fraction of $n$ | Estimated to be 0.92$^*$ |
| $v_a$ | Native ant. transport velocity of $n$ | Estimated to be 0.7 $\mu m/s^*$ |
| $v_r$ | Native ret. transport velocity of $n$ | Estimated to be 0.7 $\mu m/s^*$ |
| $\beta$ | Fragmentation rate of $m$ | Unimolecular process by which $m \rightarrow n$ |
| $\gamma$ | Aggregation rate | Bimolecular process by which $n \rightarrow m$ |
| $\delta$ | Ant. vel. enhancement factor | Effect modulated by $n$ |
| $\epsilon$ | Ret. vel. enhancement factor | Effect modulated by $m$ |

initialization and the point at which steady-state distributions are established, which typically takes a matter of months in model time with this parameterization (S1 Table). Although at these long time scales it is unlikely that the assumption of mass conservation made in our model holds, the steady-state behavior of this system is of theoretical and practical importance and explicitly modeling pathological tau recruitment would hinder the model's interpretability. At $t = 0$ (first column), the relatively high constant value of $n$ yields axonal $j_{\text{net}}$ values that are strongly positive except at the boundary between the axon proper and the AIS, where the strong concentration gradient forces flow to the left. Within the first few hours of model time (second column), the combined effects of diffusion and biased transport lead to the development of a pronounced concentration gradient within the axon proper, with a greater buildup of soluble tau towards the axon terminal due to the transport feedback. Over a period of days (third column), overall soluble tau concentrations decrease as it converts to insoluble tau through aggregation, but remain asymmetrically distributed; insoluble tau builds up within the axon proper with a concentration profile mimicking that observed for soluble tau at previous time points. At much longer time scales, aggregation causes insoluble tau to dominate, and although the anterograde-biased concentration gradient within the axon persists, that between the axon and the SD compartments becomes zero (fifth column). Although insoluble tau is strictly immobile, there is an apparent migration of insoluble tau at these later time points due to transient concentration gradients of soluble tau across between the axon and SD compartments, which emerge from the dynamic equilibrium between aggregation and fragmentation. Overall, after a period of several months, this system reaches a stable asymmetric distribution of pathological tau, with a greater buildup in the postsynaptic neuron reflecting the strong anterograde-biased transport feedback.

**Table 2. Table indicating which processes are modeled in each biological compartment.** See Methods: **Model Description** for more details.

| Compartment | Transport | Diffusion | Interconversion |
|---|---|---|---|
| Somatodendritic | No | Yes | Yes |
| Axon | Yes | Yes | Yes |
| Axon Initial Segment | No | Yes (Slow) | Yes |
| Synaptic Cleft | No | Yes (Slow) | No |

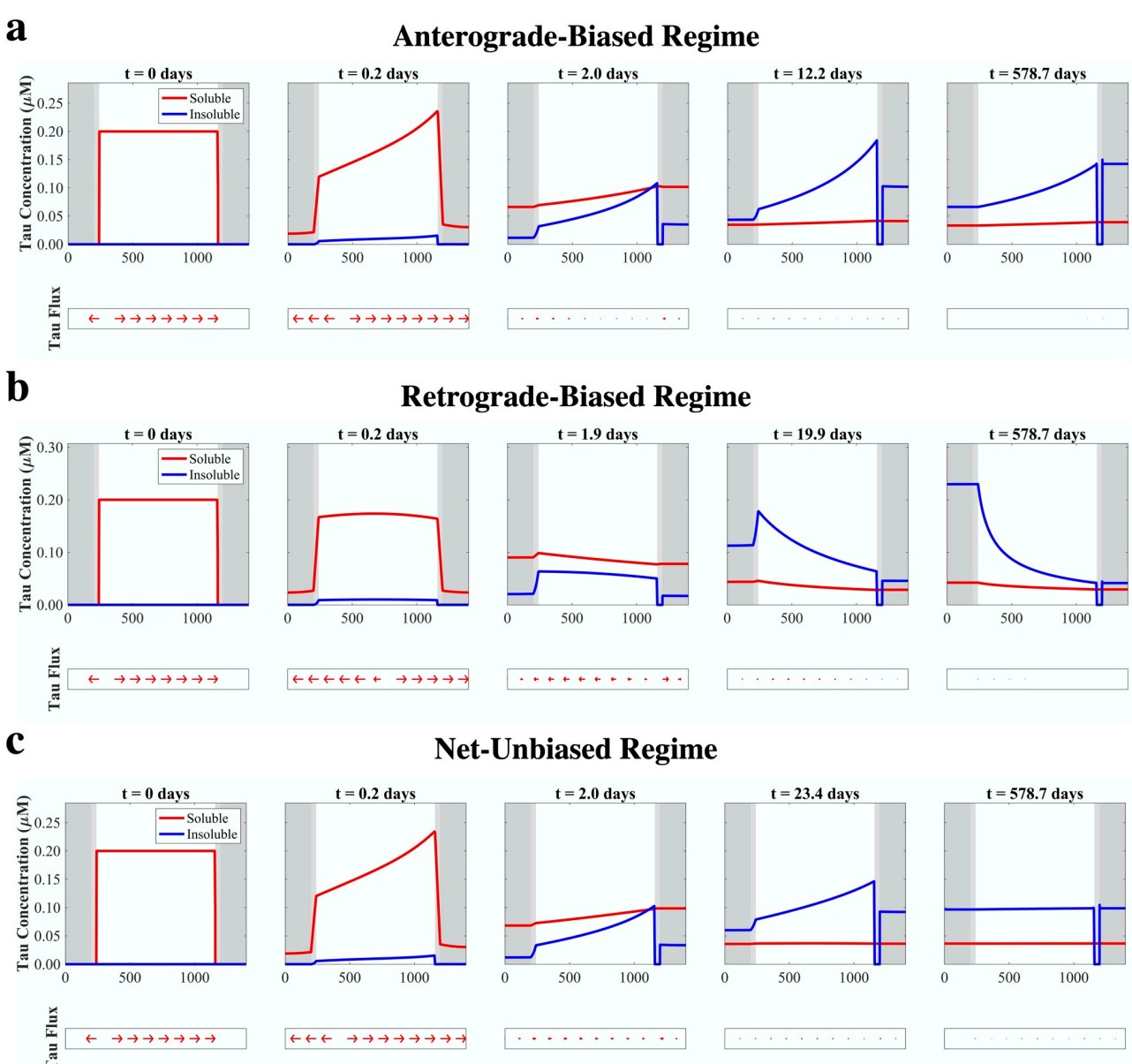

**Fig 2. Model regimes are determined by tau transport feedback parameters $\delta$ and $\epsilon$.** (a) Simulation results where $\delta = 1$ and $\epsilon = 0.01$, which leads to a strong anterograde bias that emerges within hours and persists even at longer time scales. (b) Simulation results where $\delta = 0.01$ and $\epsilon = 1$, which leads to a strong retrograde bias that only emerges at intermediate-to-late time scales. (c) Simulation results where $\delta = 1$ and $\epsilon = 0.35$, which leads to an initial anterograde bias that is counteracted at intermediate time scales, leading to a uniform distribution of tau deposition at steady state.

**2.2.2 Retrograde-biased regime.** We then simulated the condition where transport feedback is strongly biased in the opposite direction ($\epsilon \gg \delta$; Fig 2b and S2 Video). Unlike the previous condition, the lack of strong anterograde feedback parameter causes simple diffusion to dominate over the first few hours, leading to a largely symmetric concentration profile of soluble tau (second column). It is only when appreciable aggregation occurs at an intermediate time scale (third column) that an asymmetric profile is allowed to develop, as the buildup of insoluble tau leads to a net negative active transport velocity and leftward migration of soluble tau. The net deposition in the presynpatic SD compartment only grows more pronounced

over time (fourth and fifth columns), leading to an overall retrograde-biased concentration profile at steady state. Taken together with the results from the previous simulation, it is apparent that the $\delta$ and $\epsilon$ feedback parameters independently and consequentially impact how pathological tau is distributed throughout this two-neuron system.

**2.2.3 Net-unbiased regime.**   We further explored how these two modes of transport feedback can interact by setting both $\delta$ and $\epsilon$ values to be significantly greater than 0 (Fig 2c and S3 Video). Initially, as in the anterograde-dominant condition (Fig 2a), there is a pronounced asymmetric profile of soluble tau biased towards the axon terminal due to the strong positive kinesin feedback (second column). This bias persists even at longer time scales as aggregation allows insoluble tau to accumulate (third and fourth columns); however, the presence of insoluble tau begins to counteract the initial anterograde transport bias. At steady state (fifth column), the net balance between $\delta$ and $\epsilon$ leads to a nearly flat concentration profile across all compartments and roughly equal deposition between the presynaptic and postsynaptic SD compartments.

## 2.3 Somatodendritic tau deposition over time

We summarize the results of the three parameter regimes described above by plotting the mean pathological tau concentrations in each of the SD compartments across all time points of the simulation (Fig 3). Initially there is no tau of either species in the presynaptic or postsynaptic SD compartment, but at early model times ($t < 1$ day), diffusion allows soluble tau to migrate into both. In the anterograde-biased parameter regime (Fig 3a, left panel), soluble tau accumulates faster in the postsynaptic SD compartment relative to the presynaptic, which at longer time scales is converted to a persistent net accumulation of insoluble tau. Conversely, soluble tau concentrations stay at similar levels in both SD compartments up to a period of around 1 day of model time in the retrograde-biased regime (Fig 3a, center panel), where the accumulation of insoluble tau in the axonal compartment biases transport towards to the presynaptic SD compartment. Presynaptic accumulation of insoluble tau steadily increases at longer time scales while postsynaptic insoluble tau stays roughly flat, as active transport continues to push soluble tau in the retrograde direction that then forms insoluble tau through aggregation. In the net unbiased regime (Fig 3a, right panel), the resulting distributions match that of the anterograde biased regime through a period of around 1 day, until the accumulation of insoluble tau inside the axon produces a counteracting retrograde bias that leads to roughly equal concentrations of both species in each compartment. These results can be summarized by the a single "bias" metric, which we define as the difference between total postsynaptic tau and total presynaptic tau divided by total somatodendritic tau (Fig 3b, magenta dotted lines); by this convention, zero bias indicates equal deposition between SD compartments, with positive values indicating anterograde bias and negative values indicating retrograde bias. The steady-state configurations of each parameter regime are schematized in Fig 3c, exemplifying how the $\delta$ and $\epsilon$ transport parameters alone strongly influence how tau is distributed between presynaptic and postsynaptic neurons.

## 2.4 Aggregation and fragmentation rates perturb transport feedback

We more thoroughly explored the parameter dependence of the steady state by calculating the SD bias as defined above across a broad range of transport bias parameter values, keeping all other conditions the same as the previous simulations (Fig 4a and S1 Table). We found a linear manifold separating ($\delta$, $\epsilon$) pairs that result in a net postsynaptic accumulation of tau (red) from those resulting in presynaptic accumulation (blue), indicating that it is the ratio of these two parameters that ultimately governs the end state of the system. However, we also wanted to

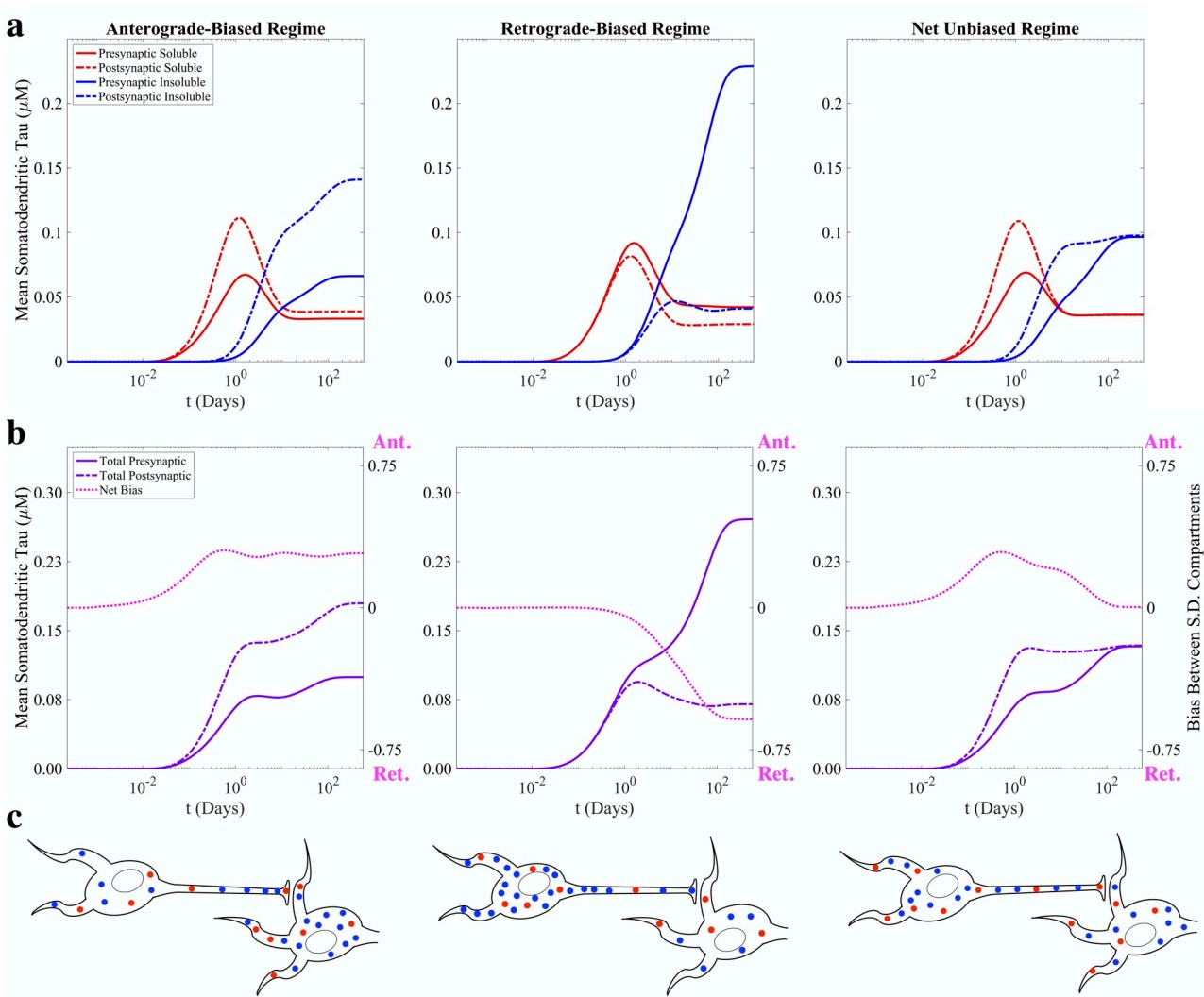

**Fig 3. Temporal profiles of somatodendritic tau deposition.** (a) Time course of mean soluble (red) and insoluble (blue) tau deposition in the presynaptic (solid lines) and postsynaptic (dashed lines) SD compartments for each of the previous simulation conditions. (b) Total presynaptic (purple solid line) and postsynaptic (purple dashed line) tau can be used to calculate a net bias at each time point (magenta dotted line; Eq 15). The balance between transport parameters $\delta$ and $\epsilon$ determines the compartment in which tau preferentially accumulates and if there is a net bias over time. (c) Schematized versions of the end configurations of the system for each parameter regime.

explore how the interconversion parameters for aggregation ($\gamma$) and fragmentation ($\beta$) influence steady-state bias, so we repeated this analysis for different values of each (Fig 4b and 4c, and S2 Fig). When $\gamma$ is doubled (Fig 4b), there is still a linear manifold of zero bias, although the slope of the line is roughly twice that of the original $\gamma$ value (Fig 4a); similarly, the slope of the zero-bias manifold is roughly halved when $\gamma$ is halved (Fig 4c). We observed a similar effect when we perturb $\beta$, albeit in the opposite direction (S2 Fig). Conversely, we found that the parameter governing the fraction of diffusing soluble tau, $f$ (Eq 4), does not affect the position of the zero-bias manifold, although it is inversely related to the intensity of the bias for any given $(\delta, \epsilon)$ pair not on the manifold (S3 Fig). An explanation for this model behavior is that the interconversion parameters influence the equilibrium between insoluble and soluble tau, and therefore undergird the relative impact of $\delta$ and $\epsilon$ on the effective transport velocity, $\nu$

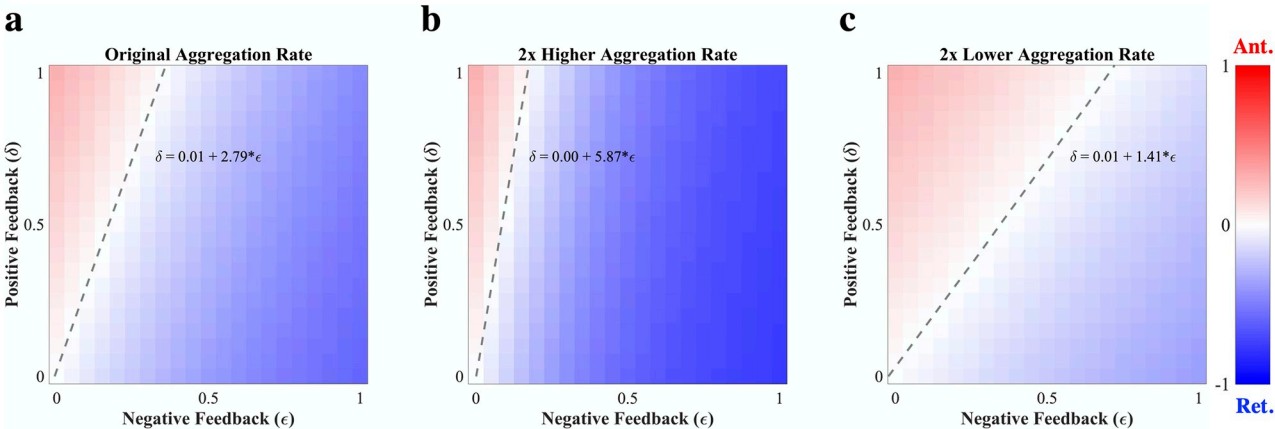

**Fig 4. Steady-state bias analysis as a function of aggregation rate.** (a) Steady-state bias (postsynaptic SD tau—presynaptic SD tau / total SD tau) across a range of $\delta$ and $\epsilon$ parameter values where all other parameter values are identical to those of the previous simulations. There is a zero-bias linear manifold that emerges, whose best-fit line has a slope of approximately 2.8. (b) Steady-state bias for the same range of $\delta$ and $\epsilon$ parameter values where aggregation rate ($\gamma$) is doubled. The linear zero-bias manifold has a slope of $\sim 5.8$, roughly twice that of the original aggregation rate. (c) Steady-state bias for the same range of $\delta$ and $\epsilon$ parameter values where $\gamma$ is halved. Here the slope of the linear manifold is $\sim 1.4$, or approximately half that of the original aggregation rate.

(Eq 1). For example, when aggregation rate is increased and there is comparatively more insoluble than soluble tau in the system, the $\epsilon$ value required to achieve the same value of $v$ is reduced, which ultimately determines how pathological tau is apportioned between SD compartments. The diffusing fraction parameter ($f$), by contrast, changes the extent to which axonal transport, as opposed to diffusion, influences the overall distribution of tau in the system and therefore affects the strength of $\epsilon$ and $\delta$ to the same extent. We conclude that, although the time course of SD tau deposition is a complex function of the interconversion and transport parameters (Fig 3), the steady state can be characterized in terms of a linear combination of these parameters.

## 2.5 Robustness to initial conditions

The previous simulations were initiated with a constant concentration of soluble pathological tau inside the axonal compartment. To explore model dependence on initial conditions, we reran the model with the same parameters as above but "seeded" either the postsynaptic or presynaptic SD compartment with a high constant concentration of insoluble tau (anterograde-biased, retrograde-biased, and net-unbiased parameter regimes shown in S4, S5 and S6 Figs, respectively); this partly mimics *in vitro* experiments where two connected neurons are kept microfluidically isolated [10]. Because only soluble tau can diffuse or be transported, the initial dose of insoluble tau must first undergo the relatively slow process of fragmentation before tau of either species can migrate to neighboring compartments, and so it takes a period of days before tau begins to build up in the opposite SD compartment. However, once steady state is established, the spatial concentration profiles of both species are indistinguishable from those yielded under the constant axonal soluble tau initial conditions (Fig 2). To more thoroughly explore the robustness of this apparent convergence to a single fixed point, we simulated 100 random initial conditions with the same total tau mass (i.e. the same spatial integral of $n$ plus $m$) as in Fig 2 and plotted the relative pairwise error between model instances as a function of time (Fig 5). In all three parameter regimes, convergence to the same fixed point is assured, despite the initial conditions themselves being quite divergent from each other. While we

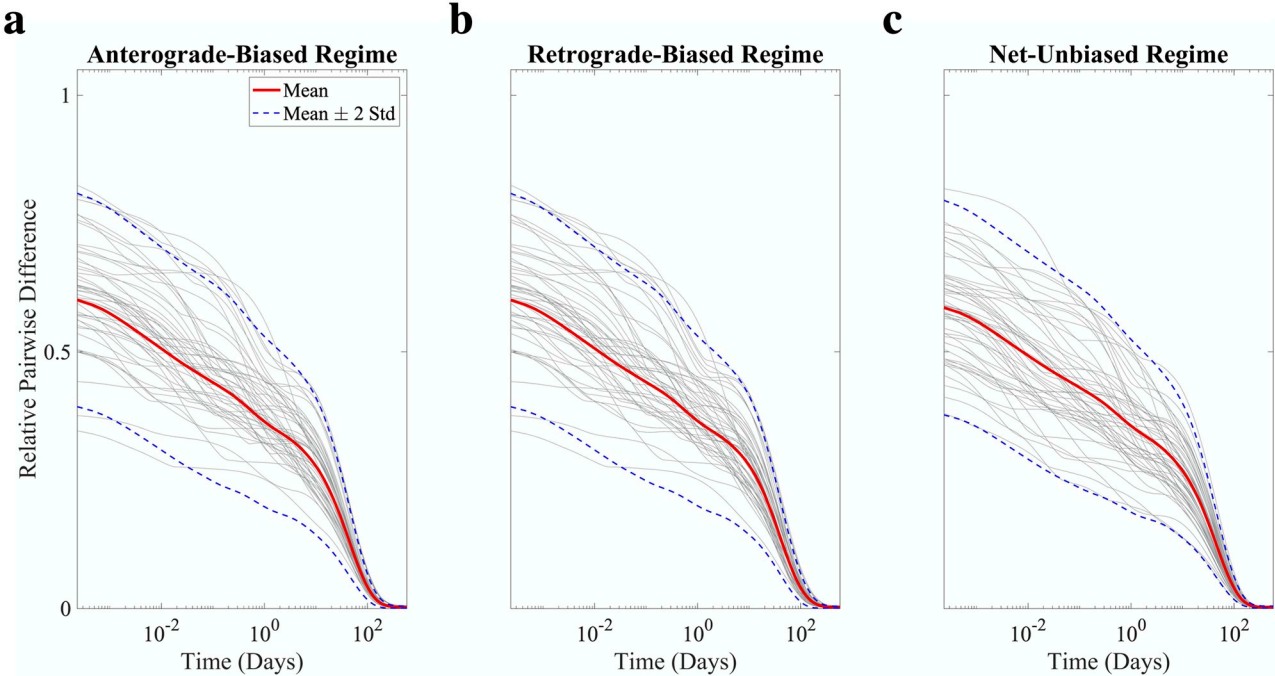

**Fig 5. Perturbation of initial conditions does not affect steady state.** We plot the relative pairwise error between model instances with randomly generated initial conditions using (a) the anterograde, (b) the retrograde, or (c) the net unbiased parameterizations (gray lines are representative sample traces). There is universal convergence at long time scales, suggesting that for these parameter values the model has a single fixed point.

cannot conclude from these simulations that this system of PDEs does not exhibit multistability at other sets of parameter values, it does suggest that with these biologically plausible conditions, there is a single steady state that is strictly dictated by model parameterization, not initial conditions.

## 2.6 Comparison to mouse tauopathy models

Although our transport feedback model is robust enough to explain a wide range of apparent directional biases in this two-neuron system, a lingering question is how applicable it is to *in vivo* models of tauopathy. We therefore parameterized our model such that the bias in SD tau deposition over time could match the apparent directionality bias observed within mouse models of tauopathy at the network level [20]. In this study, network bias was parameterized as the extent to which the spread of tau over the brain network is biased along retrograde-directed or anterograde-directed connections and its value was fit at each experimental time point using regional tau deposition data from a wide variety of mouse tauopathy models. Fig 6 shows the apparent directionality bias in several mouse models [11, 13, 36, 37] (gray data points) alongside the temporal evolution of SD bias starting from a uniform axonal concentration of soluble tau (magenta dotted lines), where we have separated mouse models based on whether tau conformers were formed in the presence of Aβ (AD-like, panel a) or not (non-AD-like, panel b). In both cases, we see good concordance ($R^2 \geq 0.4$) between transport model bias and network bias over the same time scale (S7 Fig). The tendency for an increasingly strong retrograde bias in tau propagation over time is captured by a high value of the kinesin inhibition parameter $\epsilon$. The early anterograde bias present in the non-AD-like studies also requires a high value of the kinesin rate enhancement parameter $\delta$ in contrast to the AD-like

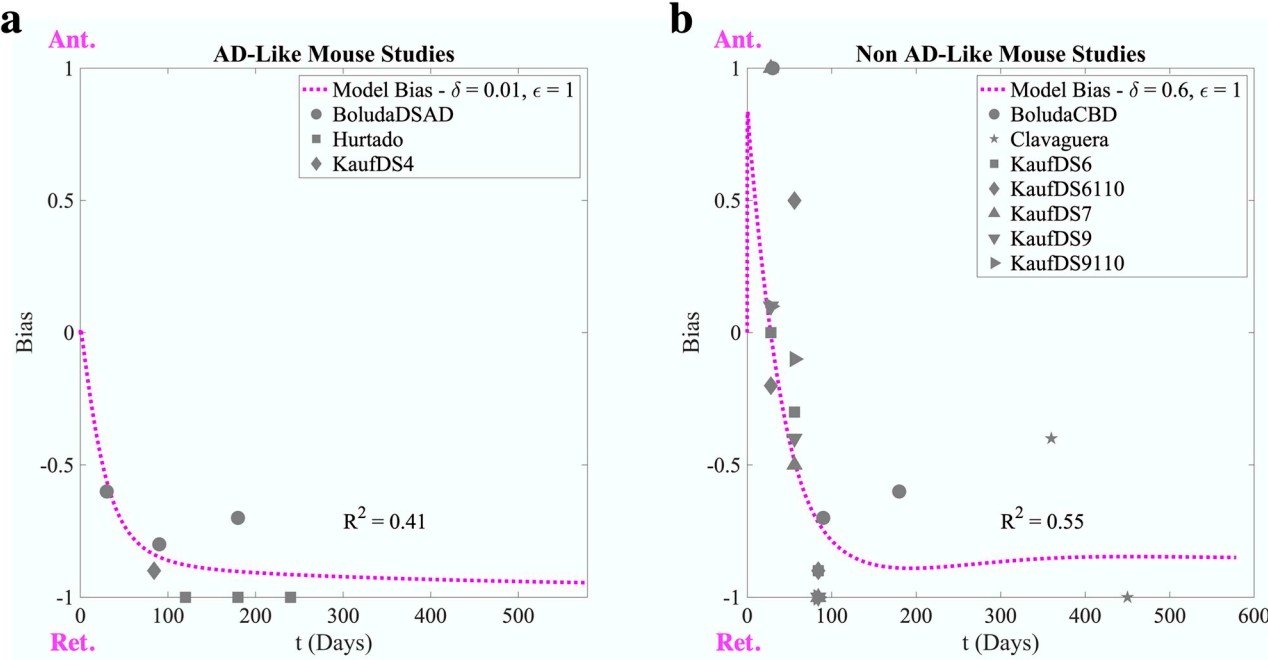

**Fig 6. Tau transport feedback recaptures the directionality of mouse tauopathy models.** (a) The AD-like mouse models explored by [20] exhibit a strong retrograde bias that becomes more pronounced over time, which can be replicated in the two-neuron system with weak anterograde-directed transport feedback (low $\delta$ relative to $\epsilon$). (b) The non-AD-like mouse models similarly have a trend towards increasing retrograde bias, although to a lesser extent than the AD-like studies and there is evidence of early anterograde bias, which is captured by fixing both $\delta$ and $\epsilon$ at high values. For all studies, we first linearly transform the bias parameter, $s$, used by [20] onto the $[-1, 1]$ scale of our SD bias estimates before plotting. Refer to S2 Table for a full parameterization of both simulations and Methods: **Analysis**. Studies included: [11, 13, 36, 37].

studies. This suggests a weaker interaction between the soluble tau assemblies formed in the presence of amyloid with kinesin than those formed in its absence. There is also a higher aggregation to fragmentation rate ratio for the AD-like studies, which similarly induces stronger and earlier retrograde biases. Given the relative paucity of data against which we compare the transport model, we cannot claim that these parameterizations uniquely fit each set of tauopathy models. Rather, our model highlights two biophysical mechanisms—aggregation to fragmentation rate ratio and differential interactions between soluble tau and kinesin—that can explain the differences between the network biases that develop in AD-like and non-AD-like mouse models. **Together, these results support axonal transport feedback as a plausible mechanistic link between differences in tau conformation and the resulting divergence in whole-brain networked spread patterns**.

## 3 Discussion

### 3.1 Transport bias as a mechanism for explaining directional bias

The present work demonstrates, for the first time, how directional bias can emerge on a microscopic level as a consequence of active transport feedback by pathological tau species. The central finding is that the model's free parameters, which include aggregation rate, fragmentation rate, and the transport feedback modifiers (Fig 1), strongly determine the spatial segregation of pathological tau over time and at steady state, with enough inherent robustness to explain both anterograde-biased spread and retrograde-biased spread. At model initiation, a combination of diffusion and anterograde-biased transport (assuming sufficiently high $\delta$) determines

how soluble tau is distributed at early time points (Fig 2, first two columns). At intermediate and late time points, where aggregation has allowed sufficient concentrations of insoluble tau to accumulate in the axon, negative feedback proportional to the value of $\epsilon$ drives tau back towards the presynaptic SD compartment (Fig 2, final three columns). Indeed, the temporal evolution of SD tau concentrations exhibits three phases of behavior for all three parameter regimes: (i) an initial rise in soluble tau (bias between compartments influenced by positive transport feedback); (ii) a fall in SD soluble tau coupled with a rise in SD insoluble tau as aggregation occurs; (iii) final re-equilibration (Fig 3a). These phases are a direct consequence of phenomena operating under dramatically different time scales, with transport and diffusion being faster processes than interconversion and inter-compartmental spread. Despite minor differences in trajectories over time, this system of coupled nonlinear PDEs has a single fixed point over a broad range of initial conditions and parameter regimes (Fig 5), which suggests that steady-state bias can be characterized, as suggested by Fig 4, S2 and S3 Figs, as a well-behaved function of the internal parameters. Our results indicate that even a relatively simple, concentration-dependent feedback mechanism (Eq 1) is sufficient to drive a wide range of model behaviors.

## 3.2 Comparison with *in vivo* mouse models

Our model behavior shows good agreement ($R^2 \geq 0.4$) with *in vivo* models of tauopathic disease at the regional level: for both AD-like and non-AD-like mouse models, we can parameterize our transport feedback model to produce biases in somatodendritic tau deposition that match the longitudinal trends in network spread bias (Fig 6 and S7 Fig) [20]. Specifically, we show that a parameter regime with high $\epsilon$ relative to $\delta$ can reproduce directional bias trends in AD-like mouse models, while for the non-AD-like models we better capture bias with higher $\delta$ (S2 Table). The model regimes chosen also exhibit different balances of aggregation and fragmentation parameters, with a relatively higher aggregation rate in AD-like mouse models; this tends to further strengthen the bias imposed by $\epsilon$ relative to $\delta$ (Fig 4 and S2 Fig). These two effects act in concert to produce retrograde directional biases in AD-like tau models that are stronger and have earlier onsets than those observed in non-AD-like models. Therefore, we suggest that an amyloidogenic microenvironment may produce tau conformers that have higher rates of aggregation relative to fragmentation and/or differential interactivity with kinesin. amyloid-β has long been considered to be the best biomarker available for distinguishing AD from other dementias, and more recently positive amyloid status has been shown to be strongly related to the presence of the p181 tau phosphoepitope in patients [16, 18]. Although the indirect and direct interactions between amyloid-β and tau are complex and the subject of intense scrutiny, our work suggests that tau conformers cultured in the presence of amyloid-β may have specific properties impacting axonal transport that merit further exploration.

## 3.3 Limitations

There are several important limitations to the present work. Most notably, due to the paucity of data measuring intra-axonal tau in the disease state, it is challenging to directly fit our model parameters or validate our findings. Although *in vitro* time-lapse microscopy studies of tau transport kinetics have been conducted with both tau in its native conformation and pseudophosphorylated variants [28, 29], there have been no investigations of pathological tau dynamics that determine concentration profiles over time in single neurons. We also note that, because there are few studies for which direct regional quantification of mouse tau pathology is available and each study reports at most four time points, our comparison between model bias and network directional bias suffers from limited statistical power. Our

model's single spatial dimension does not represent the complex geometry of the SD compartments, preventing us from accurately modeling missorting between the dendritic spines and the rest of the soma; accordingly, we only compare the mean concentrations in these compartments (Figs 3 and 4). The AIS and SC are modeled as identical diffusion-limiting barriers save for the lack of interconversion in the SC, which is implausible given their true dimensions and biological properties. Firstly, it is unlikely that these two barriers are equally impermeable to all tau species, as we encoded here. It is likewise inaccurate to assume that it is a purely diffusive process that facilitates tau migration across the cell membranes on either side of the SC or through the structural mesh of the AIS, but our choice was motivated by our expectation that transport rate across these boundaries should be roughly proportional to the concentration difference across them. An intriguing possibility for future modeling work is to incorporate a concentration-dependent permeability; for instance, tau missorting into the SD compartment is a consequence of AIS breakdown, which suggests a dynamic $\lambda$ [25]. Finally, we entirely neglect the conversion of healthy tau to pathological soluble tau, pathological tau clearance, and the spread of tau outside of the two-neuron system from our model since these effects would hinder model interpretability.

## 3.4 Applications to other neurodegenerative diseases

Axonal transport feedback may also play a role in disorders such as Parkinson's disease (PD) and amyotrophic lateral sclerosis (ALS). Experimental and clinical studies have reported prion like spreading of misfolded protein assemblies along the white matter tracts of the brain in these disorders, and mathematical models successfully captured key features of their neuropathology [38–41]. Because migration along axonal fibers is necessary for infiltration into areas of the brain beyond regions where pathology initiates, how the axonal active transport machinery interacts with misfolded assemblies of $\alpha$-synuclein, TDP-43, and SOD1 remains an open question. There is evidence that $\alpha$-synuclein interacts with both kinesin-1 and dynein [32], and down-regulates them at high levels [42]. It is therefore possible that $\alpha$-synuclein, as we have explored here with tau, perturbs the balance of axonal transport in a conformer-specific fashion, potentially explaining some of the clinical heterogeneity observed in PD [43, 44]. Similarly, axonal transport defects in both directions are a consistent and early feature of ALS [45], with evidence of directionally biased spread of SOD1 in a transgenic mouse model [46]. Although evidence of the dual feedback mechanism for pathological tau explored here is less clear for these diseases, incorporating the effects of transport defects could lead to better predictive models for non-tauopathic proteinopathies.

## 3.5 Integrating axonal transport at a network level

A necessary extension to the current work will be to scale up the two-neuron system of PDEs explored here to a network of regions, which will facilitate parameter fitting and allow us to directly compare our model output to histopathological data. Conformer-specific parameters such as aggregation rate and the transport feedback modifiers should be *globally invariant* to avoid overfitting; that is, they should not depend on the compartment in which the pathological tau resides. Then, using inter-regional connectivity to weight pathology spread, we can directly simulate how tau transport feedback effects spread in a macroscopic sense [20, 47–50]. Explaining directionally biased spread in mouse models of tauopathy is the most natural immediate fit to the current work, as there are many models for which tau pathology has been extensively detailed [11, 13, 36, 37] and the mouse mesoscale connectivity atlas [51] separates efferents from afferents, giving "anterograde" and "retrograde" a precise anatomical meaning. Application to humans is more challenging given that diffusion tensor imaging cannot

typically determine the orientation of white matter tracts. However, a hybrid-species connectome that encodes directionality for evolutionarily conserved connections has been successfully used to model the spread of neuropathology in progressive supranuclear palsy [52].

## 3.6 Summary

We have demonstrated that transport feedback is a simple and sufficient mechanism to explain a broad range of directional biases in tau spread in a two-neuron system. Although much about tau biology in healthy and pathological states remains poorly understood, and many other factors besides tau conformational status may contribute to the progression of tauopathic diseases, it is increasingly clear that there are disease-specific tau conformers that have distinct patterns of spread. We anticipate that differences between tau conformers in their endogenous ability to regulate kinesin, in addition to disparities in aggregation and fragmentation rates (among others), is a parsimonious mechanism that, coupled with networked, trans-synaptic spread, drives not only heterogeneity within a given tauopathy but also the distinct pathophysiologies of different tauopathies.

## 4 Methods

### 4.1 Model description

We use a coupled system of partial differential equations (PDEs) to model the concentrations of two species of pathological tau: soluble $n(x, t)$, which can travel via *diffusion* and *active transport* processes (the net balance of which determines the effective transport of pathological tau), and insoluble tau $m(x, t)$, which is immobile. These two species interconvert through two opposing processes: (i) *aggregation*, a bimolecular reaction between one unit of $n$ and one unit of either $n$ or $m$ to form $m$; and (ii) *fragmentation*, a unimolecular reaction by which a unit of $m$ breaks apart to form a unit of $n$. We did not model healthy tau concentrations or *recruitment*, the process by which tau loses its native conformation via aberrant post-translational modifications such as hyperphosphorylation, fragmentation, and acetylation to adopt a pathological, prion-like conformation. As such, the total mass in the system is conserved, since fragmentation and aggregation act symmetrically on $n$ and $m$. The unique feature of this model is that we incorporate the feedback these two species provide on kinesin, with soluble tau enhancing kinesin processivity [28–30] and insoluble tau inhibiting it [31]. Mathematically, we propose the following relationship for the effective transport velocity:

$$v(n, m) \equiv v_a \cdot (1 + \delta n)(1 - \epsilon m) - v_r ,\tag{1}$$

where $v_a$ and $v_r$ are the baseline anterograde and retrograde velocities of tau, respectively, $\delta$ is a nonnegative parameter governing the enhancement of kinesin processivity in response to soluble pathological tau, and $\epsilon$ is a nonnegative parameter governing the reduction of kinesin processivity in response to insoluble pathological tau. In this expression we have assumed that on the time scale of the simulation (days), the characteristic "start-and-stop" dynamics of molecular motors, where motor binding and unbinding events happen within fractions of a second [53], can be subsumed into effective velocity parameters. We anticipate that the separation of time scales between characteristically "slow" processes such as aggregation and fragmentation and the "fast" dynamics of motor-microtubule interactions allows us to time-average the latter without a loss of generality. By convention, positive $v$ implies net transport from presynaptic to postsynaptic neuron (i.e. anterograde transport). The various compartments of this two-neuron system necessarily have different properties that should be modeled distinctly, and we

discuss the particulars of each compartment below. Refer to Table 1 and the text below for a complete description of the parameterization of the model.

**4.1.1 Axonal compartment.** Given the axon's high aspect ratio, which restricts movement of $n$ parallel to its long axis whether it is diffusing or being actively transported by molecular motors, we collapse the three-dimensional geometry of this system onto one dimension [35]. First we use the following expressions for diffusive flux ($j_{\mathrm{diff}}$) and transport flux ($j_{\mathrm{active}}$) of $n$:

$$j_{\mathrm{diff}}(n_x) \quad = \quad -D_n \cdot n_x \,, \tag{2}$$

$$j_{\mathrm{active}}(n, m) \quad = \quad v(n, m) \cdot n \,, \tag{3}$$

where parameter $D_n$ is the diffusivity of soluble pathological tau, $n_x$ denotes the partial derivative of $n$ with respect to $x$, and $v(n, m)$ is given by Eq 1. Under the assumptions that i) active transport and diffusion are mutually exclusive processes and ii) the processes of attachment and detachment from molecular motors and microtubules are sufficiently "fast" on the simulation time scale, we define the net flux of $n$, $j_{\mathrm{net}}$, in the following way:

$$j_{\mathrm{net}}(n, m, n_x) \equiv f \cdot j_{\mathrm{diff}}(n_x) + (1 - f) \cdot j_{\mathrm{active}}(n, m) \,, \tag{4}$$

where $f$ is the average fraction of soluble pathological tau that is undergoing diffusion at any given time. While we estimate $f$ based on measurements made by Konzack *et al.* [34] using pseudophosphorylated tau to mimic disease, we recognize that there is considerable uncertainty in this parameter and therefore demonstrate how it influences the model (see Results: **Parameter Dependence of Steady-State Bias**; S3 Fig). It is also worth noting that the diffusivity value measured by Konzack *et al.* [34] in cultured rodent neurons of 11 $\mu$m$^2$/s is the effective diffusivity rate utilized in $j_{\mathrm{net}}$, since $D_n \cdot f = 12 \mu\mathrm{m}^2/\mathrm{s} \cdot 0.92 = 11 \mu\mathrm{m}^2/\mathrm{s}$. The dynamics of interconversion of $n$ and $m$ as described above are governed by the following expression:

$$\Gamma(n, m) \equiv \beta m - \gamma n(n + m) \,, \tag{5}$$

where we subsume the intricate dynamics of fragmentation and aggregation into two parameters, $\beta$ and $\gamma$. Combining these definitions, we arrive at the PDEs governing the dynamics of $n$ and $m$ inside the axon:

$$\frac{\partial n_{\mathrm{axon}}}{\partial t} \quad = \quad -\frac{\partial}{\partial x} j_{\mathrm{net}}(n, m, n_x) + \Gamma(n, m) \,, \tag{6}$$

$$\frac{\partial m_{\mathrm{axon}}}{\partial t} \quad = \quad -\Gamma(n, m) \,, \tag{7}$$

where $j_{\mathrm{net}}$ is given by Eq 4 and $\Gamma$ is given by Eq 5.

**4.1.2 Somatodendritic compartments.** Our two-neuron system incorporates two somatodendritic (SD) compartments, corresponding to the presynaptic and postsynaptic cell bodies, which are modeled identically. For the purposes of this model, we focus on the overall amounts of SD tau—particularly the differences between presynaptic and postsynaptic neurons—that result from the tau dynamics within the axon. Although a one-dimensional model cannot be used to precisely model the spatial distributions of tau in these compartments, which have complex, three-dimensional geometries, a one-dimensional approximation is sufficient to capture the mean somatodendritic tau concentrations. We therefore greatly simplify the system by modeling these compartments as one-dimensional segments 200$\mu$m in length. The dynamics of the SD compartments only differ from the axonal compartment regarding the active transport of $n$. We do not expect transport to be more efficient than diffusion in the

soma, and transport in the dendrites is both distinct from that in the axons and beyond the scope of our model. Therefore, the governing equations are given by:

$$\frac{\partial n_{\text{SD}}}{\partial t} \quad = \quad -\frac{\partial}{\partial x} j_{\text{diff}}(n_x) + \Gamma(n, m)\,, \tag{8}$$

$$\frac{\partial m_{\text{SD}}}{\partial t} \quad = \quad -\Gamma(n, m)\,. \tag{9}$$

**4.1.3 Axon initial segment.**   The axon initial segment (AIS) constitutes the first 20–60 $\mu$m of the axonal length beyond the hillock, and has several features that distinguish it from the axon proper. The intricate mesh of structural proteins occupying the AIS serves as a barrier that restricts the free intermixing of axoplasm and cytoplasm components. For tau in particular, it serves as a one-way molecular sieve that allows tau to migrate from the SD compartment to the axon, but not the reverse [54]. Indeed, an early feature of AD pathology is the missorting of tau in the SD compartment following a compromise of AIS integrity [23, 25, 55]. We model the AIS as having a length of 40 $\mu$m with dynamics similar to the SD compartments, with the exception that the effective diffusivity of $n$ is modified by the free parameter $\lambda$, whose value is fixed at a value much less than one:

$$\frac{\partial n_{\text{AIS}}}{\partial t} \quad = \quad -\lambda \cdot \frac{\partial}{\partial x} j_{\text{diff}}(n_x) + \Gamma(n, m)\,, \tag{10}$$

$$\frac{\partial m_{\text{AIS}}}{\partial t} \quad = \quad -\Gamma(n, m)\,. \tag{11}$$

In this way, the AIS serves as a fixed diffusion-based barrier over the course of the simulation.

**4.1.4 Synaptic cleft.**   The synaptic cleft (SC) is the narrow extracellular space between the axon terminus of the presynaptic neuron and the dendrite of the postsynaptic neuron, through which the neurotransmitters that mediate the flow of information carried by action potentials are released and taken up [56]. Trans-synaptic spread of pathological tau is thought to be the dominant mechanism by which tau migrates between neurons in the brain [11, 23], although the precise molecular mechanisms by which it traverses the SC remain the subject of intense scrutiny [57, 58]. For our purposes, the SC acts as a barrier that restricts free diffusion of $n$ between presynaptic and postsynaptic neurons, and, in the absence of a more precise way of describing its dynamics, we chose to model it identically to the AIS, with the exception that we do not allow $m$ to accumulate or $n$ to aggregate, given the true spatial constraints of the synapse:

$$\frac{\partial n_{\text{SC}}}{\partial t} \quad = \quad -\lambda \cdot \frac{\partial}{\partial x} j_{\text{diff}}(n_x)\,, \tag{12}$$

$$\frac{\partial m_{\text{SC}}}{\partial t} \quad = \quad 0\,. \tag{13}$$

We have chosen to use the same parameter $\lambda$, which governs the effective diffusivity of $n$, for both the AIS and the SC for two reasons. One, we lack quantitative experimental studies from which these effective permeabilities can be ascertained, although these barriers are likely to be differentially permeable to pathological tau in a neuron-subtype-specific and conformer-specific way. Two, from a modeling perspective, this allows us to cleanly explore the effects of

the axonal transport parameters $\delta$ and $\epsilon$ given the lack of mechanistic knowledge about how pathological tau migrates out of the axonal compartment.

## 4.2 Numerical implementation

We utilize the MATLAB parabolic and elliptical PDE solver `pdepe`, which efficiently discretizes a system of one-dimensional PDEs over a specified spatial mesh and solves the resulting system of differential algebraic equations (DAEs) using `ode15s` [59].

**4.2.1 Initial conditions.** Aside from analyses where we directly explored the effect of model initialization (see Results: **Robustness to Initial Conditions**; Fig 5), for all simulation results presented herein, we specify a constant $n$ concentration of 0.2 $\mu$M within the axonal compartment and 0 elsewhere at $t = 0$. We also stipulate that there are no insoluble aggregates at the start of the simulation. This condition simulates the initiation of tau pathology in the axon and its resulting spread to neighboring, otherwise unaffected neuron cell bodies, as is hypothesized to occur in early tauopathic disease [23, 60]. Although the "true" initial configuration of the system would be the healthy state with no $n$ or $m$ anywhere, with $n$ gradually added to the system through the conversion of healthy tau in the axon (i.e. recruitment), this process depends upon poorly characterized time-dependent concentration profiles of healthy tau and numerous other species (e.g. axonal kinases and phosphatases) and goes beyond the scope of the present work. Since the focus of our model is how different pathological tau species, once formed, may have apparent directional biases as a function of their propensity to aggregate, fragment, and regulate axonal transport, we believe that our choice of initial conditions sufficiently describes an early state of pathology in tauopathic diseases.

**4.2.2 Spatial mask and boundary conditions.** We modify the dynamics across compartments using discontinuous, binary *spatial masks* and simulate the whole system in one call of `pdepe`. These masks allow us to turn "on" and "off" processes such as active transport and interconversion depending on $x$ position. This methodological choice allows us to take full advantage of the efficient `pdepe` algorithm and grants us the flexibility to define 1-D geometries of arbitrary complexity. Additionally, it eschews the need to encode complicated boundary conditions for each compartment and then couple them at each time step, which would introduce significant inaccuracies. We use zero-flux Neumann boundary conditions at $x = 0$ (left boundary of the presynaptic SD compartment) and $x = L_{\text{total}}$ (right boundary of the postsynaptic SD compartment), ensuring that mass in the system is conserved at every time point. While these boundaries are not impermeable in the open biological system of the brain on the time scale of the simulation, we chose not to include leakage because it would reduce model interpretability and cannot be accurately simulated without also explicitly modeling the concentration profiles in all neighboring cells. In order to minimize inaccuracies due to the lack of differentiability introduced by the spatial mask while maintaining short computation times per simulation, we specify an inhomogeneous spatial mesh that is very fine within 10 $\mu$m of each compartment boundary on either side and coarser elsewhere.

**4.2.3 Parameterization.** For the effective transport parameters that were estimated in a comparable *in vitro* system, we rely upon the experimental work of Konzack *et al.* in cultured rodent neurons [34]. The interconversion parameters in S1 Table, which are more challenging to estimate with high precision in a cellular context, were tuned to yield biologically plausible rates of insoluble tau accumulation. The transport feedback parameters, whose impact on model behavior is the primary focus of the present work, were each varied within a range of 0 to 1.

### 4.3 Analysis

**4.3.1 Assessment of steady state.** In lieu of directly solving a boundary value problem to find steady-state distributions, which poses a challenge given the discontinuities in the spatial derivatives between biological compartments, we found steady-state distributions by simulating the model equations out to sufficiently long times where the distributions no longer appreciably change. We make this assessment precise by calculating the approximate rate of change of the distributions, $\frac{\Delta \tau}{\Delta t}(t)$:

$$\frac{\Delta \tau}{\Delta t}(t) = \frac{|n(t) - n(t-1)|_1 + |m(t) - m(t-1)|_1}{|n(t)|_1 + |m(t)|_1} / \Delta t . \tag{14}$$

We plot this rate of change over time and show that it converges to effectively zero at the end time of our simulations (S1 Fig).

**4.3.2 Directional bias.** We assess the bias in tau deposition over time, $B(t)$, as a simple function of the mean concentrations in the two somatodendritic compartments as follows:

$$B(t) \equiv \frac{(\bar{n}_{\text{post}}(t) + \bar{m}_{\text{post}}(t)) - (\bar{n}_{\text{pre}}(t) + \bar{m}_{\text{pre}}(t))}{\bar{n}_{\text{post}}(t) + \bar{m}_{\text{post}}(t) + \bar{n}_{\text{pre}}(t) + \bar{m}_{\text{pre}}(t)} , \tag{15}$$

where $\bar{n}_{\text{pre}}(t)$, $\bar{m}_{\text{pre}}(t)$, $\bar{n}_{\text{post}}(t)$, and $\bar{m}_{\text{post}}(t)$ are the mean soluble and insoluble tau concentrations in the presynaptic and postsynaptic SD compartments at time $t$, respectively. In this way, a bias value of 1 represents purely anterograde-biased deposition in the postsynaptic SD compartment, a bias value of -1 represents purely retrograde-biased deposition in the presynaptic SD compartment, and a bias value of 0 indicates equal deposition in both compartments.

**4.3.3 Bias manifold calculation.** The topology of the $\delta$-$\epsilon$ parameter space at steady state is of keen interest and we sought to characterize the zero-bias manifold: the boundary between parameter combinations that result in anterograde-biased deposition and those that result in retrograde-biased deposition. To calculate these linear manifolds, we first fit a cubic polynomial to the steady-state bias values, $B(t_{SS})$ (Eq 15), with respect to $\epsilon$ for each unique $\delta$ value. The real root of each of these polynomials in the [0, 1] domain represents the $(\delta^*, \epsilon^*)$ pair at which the model would exhibit zero net bias at steady state. We then fit a line to these $(\delta^*, \epsilon^*)$ values to get a linear parameterization of the zero-bias manifold.

**4.3.4 Random initialization.** For each model instance, we set the concentration of $n$ and $m$ in each somatodendritic compartment and at ten equally spaced points within the axonal compartment to be a random number between 0 and 1. We then linearly interpolate between these values across the full spatial domain and normalize both distributions such that the total mass is the same as the original initial condition (see Methods: **Initial Conditions**).

**4.3.5 Parameter fitting.** We varied $\beta$, $\gamma$, $\delta$, $\epsilon$, and $f$ within reasonable ranges ($[1 \times 10^{-8}, 1 \times 10^{-5}]$ for $\beta$ and $\gamma$; [0, 1] for $\delta$ and $\epsilon$; and [0.5,1] for $f$) to find parameterizations with good fits to network bias data so that we could demonstrate that the model can plausibly recreate the time dependence of directional bias. The parameterizations in S2 Table do not uniquely identify the models of best fit; however, the differences between the two sets of parameters remain (e.g. higher $\gamma$ to $\beta$ ratio for AD-like mouse models) consistent for other parameterizations with comparable fits. We chose these specific parameterizations because they achieve excellent concordance with network bias and differ only in three parameters ($\beta$, $\gamma$, and $\delta$).

## Supporting information

**S1 Table. Table of all parameter values used in the simulations shown in Figs 2, 3 and 5; see Methods for a full description of each parameter.** Values marked with an asterisk were estimated from Konzack *et al.* [34].
(XLSX)

**S2 Table. Table of parameter values used in the simulations shown in Fig 6; unlisted parameters have identical values as in S1 Table.**
(XLSX)

**S1 Fig. Convergence of individual simulations to steady state.** The rate of change of tau distributions (Eq 14) for the three parameter regimes explored in the main text (S1 Table) approaches zero within the time scale of the simulation. Notably, at mid-to-late model times, convergence is well approximated by an exponential, with similar rates of decay for all three parameter regimes (right panels).
(TIF)

**S2 Fig. Steady-state analysis as a function of fragmentation rate β.** (a) Steady-state bias (postsynaptic SD tau—presynaptic SD tau / total SD tau; see Eq 15) across a range of $\delta$ and $\epsilon$ parameter values where all other parameter values are identical to those of the previous simulations. There is a zero-bias linear manifold that emerges, whose best-fit line has a slope of approximately 2.8. (b) Steady-state bias for the same range of $\delta$ and $\epsilon$ parameter values where aggregation rate ($\beta$) is doubled. The linear zero-bias manifold has a slope of $\sim 1.4$, roughly half that of the original fragmentation rate. (c) Steady-state bias for the same range of $\delta$ and $\epsilon$ parameter values where $\beta$ is halved. Here the slope of the linear manifold is $\sim 5.8$, or approximately twice that of the original fragmentation rate.
(TIF)

**S3 Fig. Steady-state analysis as a function of diffusive tau fraction *f*.** The overall strength of the net bias at any given pair of $\delta$ and $\epsilon$ values is inversely proportional to *f*, but it does not affect the line of zero bias (dashed lines), in contrast to $\gamma$ and $\beta$.
(TIF)

**S4 Fig. Somatodendritic seeding, anterograde-biased parameter values.** Seeding in either SD compartment converges to the same steady state as the axon-only initial condition in the anterograde-biased parameter regime (Fig 2a). Parameter values used identical to those in S1 Table.
(TIF)

**S5 Fig. Somatodendritic seeding, retrograde-biased parameter values.** Seeding in either SD compartment converges to the same steady state as the axon-only initial condition in the retrograde-biased parameter regime (Fig 2b). Parameter values used identical to those in S1 Table.
(TIF)

**S6 Fig. Somatodendritic seeding, net-unbiased bias parameter values.** Seeding in either SD compartment converges to the same steady state as the axon-only initial condition in the net-unbiased parameter regime (Fig 2c). Parameter values used identical to those in S1 Table.
(TIF)

**S7 Fig. Transport model bias versus mouse network bias scatterplot.** Scatterplot showing the relationship between mouse network bias across several tauopathy studies and the bias

predicted by the transport model at equivalent time points. The adjusted $R^2$ values for each set of studies are listed in the legend.
(TIF)

**S1 Video. Video of the complete simulation summarized in Fig 2a.**
(MP4)

**S2 Video. Video of the complete simulation summarized in Fig 2b.**
(MP4)

**S3 Video. Video of the complete simulation summarized in Fig 2c.**
(MP4)

## Acknowledgments

We thank Dr. Daniel Gardner for his helpful suggestions on revising the manuscript. We also thank Dr. Yi Wang for his insight and his critical support of this project.

## Author Contributions

**Conceptualization:** Justin Torok, Pedro D. Maia, Ashish Raj.

**Data curation:** Christopher Mezias.

**Formal analysis:** Justin Torok, Pedro D. Maia, Parul Verma.

**Funding acquisition:** Ashish Raj.

**Methodology:** Justin Torok, Pedro D. Maia, Parul Verma.

**Project administration:** Ashish Raj.

**Software:** Justin Torok.

**Supervision:** Ashish Raj.

**Validation:** Christopher Mezias.

**Visualization:** Justin Torok.

**Writing – original draft:** Justin Torok.

**Writing – review & editing:** Justin Torok, Pedro D. Maia, Parul Verma, Christopher Mezias, Ashish Raj.

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
