## [Decision Letter · Decision Letter 0]

11 May 2021

Dear Mr. Torok,

Thank you very much for submitting your manuscript "Emergence of directional bias in tau deposition from axonal transport dynamics" for consideration at PLOS Computational Biology.

As with all papers reviewed by the journal, your manuscript was reviewed by members of the editorial board and by several independent reviewers. In light of the reviews (below this email), we would like to invite the resubmission of a significantly-revised version that takes into account the reviewers' comments.

We cannot make any decision about publication until we have seen the revised manuscript and your response to the reviewers' comments. Your revised manuscript is also likely to be sent to reviewers for further evaluation.

Sincerely,

Michele Migliore

Associate Editor

PLOS Computational Biology

Kim Blackwell

Deputy Editor

PLOS Computational Biology

Reviewer's Responses to Questions

**Comments to the Authors:**

Reviewer #1: I: Overview:

Torok and colleagues develop a mathematical model of the spatiotemporal evolution of soluble and insoluble tau species to study if changes in axonal transport based on tau species conformer could explain differential behavior observed in tauopathies based on amyloid status.

The mathematical model is of a one-dimensional interval taken to model two neurons, but represents all the critical mechanisms in an elegant way. The results in the manuscript are well explained in words, but the presentation in the Figures was less clear and more confusing than necessary. This work has strong contributions to the field, below I offer comments which if addressed will make its particular results more clear to understand.

II: Major Comments

(1) The critical figure in this paper, Figure 1, could be improved to make it more clear which processes occur happening in which compartments. Once I read the model description it made sense, but I think with a few more additions, it could be really clear just by looking at the figure. For example, Figure 1 seems to indicate that diffusion, active transport and interconversion are only happening in the axon.

(2) The authors assume conservation of mass in this system. This, of course, makes sense for several reasons (as they write in Section 4.2.2: mathematical simplicity and because they are modeling things on a small time scale). However, in Figure 2 they show results out to 500+ days.

I had difficulty interpreting this late time result in the context of the biology. Is this still a time-scale that we would expect conservation? Was the key purpose in this late time point to simply show that the system reaches a steady-state distribution? If so, I would forgo this to the modeling section or state explicitly.

In addition, I looked for a practical definition of the Tau Flux quantity that was at the bottom of the Figure. (See point 4 below)

(3) I had a number of problems interpreting Figure 3:

(i) The bias (magenta line) in Figure 3 is confusing. This is partly because it's not adequately defined in the caption. But I remained confused by it's definition in the text of the manuscript.

Part of this is because I'd wanted to see an equation in terms of the model variables (and of course the model occurs in a later section).

Even though it would make the figure more complicated, I would have preferred to see (rather than the bias) which is somewhat difficult to interpret, the total presynaptic tau (soluble + insoluble) and postsynaptic tau (soluble + insoluble) plotted as well.

The bias then would be directly visible and clear (rather than rely on this more indirect assessment)

(ii) The figure shown in panel (b), while informative, is confusing in the present context because this is the first presentation of neurons in this manuscript!

Since not all readers of PloS Computational Biology will be familiar with neurons (and what is meant by the synapse) it would actually have been helpful to include a schematic version of a neuron in Figure 1. Or at the very minimum - label the regions in Figure 3.

(4) Figure 4 too was challenging to understand precisely. Overall the behavior shown in the plots made sense as did the authors' interpretation of it.

(i) I could not determine the precise quantity that was plotted. I suspect that this is in fact the bias quantity in Figure 3 and perhaps related to Tau Flux in Figure 2. Again, a specific equation for SD bias would have been helpful.

(ii) Finally, since the authors do not have explicit steady-state solutions to this model, I presume this bias quantity was calculated numerically at some future time?

(iii) I presume that these lines were numerically fit? If so at least a sentence about how would be helpful.

(5) While the present scope of the study sought to a numerical analysis of this model, as a mathematician I am intrigued by the possibility of analytical results for this model.

The model presented is globally attracting, and it seems that this steady-state distribution might be possible to determine analytically at least for some regimes/conditions.

Finally, the authors (Figure 5) refer to a "fixed-point". That is more commonly used for ODEs. Since the authors are using a PDE, the term "steady-state distribution" or "steady-state density" would be more appropriate.

III: Minor Comments & Typos

(1) In Table 1, I'd like to have a little more clarity of where the estimated parameters came from. The authors cite [34], but something such as, "Estimated by fitting a model to mouse data", "Estimated from in vitro assays". Something to give your readers the understanding of where the values came from.

(2) Page 3/4: Change "biological compartment it resides" to "biological

compartment in which it resides"

(3) Page 7: Change "by total somatodentric" to "by total somatodendritic

(4) Page 9: Change "with these biological plausible conditions" to "with these

biologically plausible conditions"

(5) Page 14 in the Model section has a number of Typos:

(i) Change: "demonstrate how the it influences model" to "demonstrate how it influences the model"

(ii) I recommend replacing the following text: "by modeling these compartments with a 200 m one-dimensional lengths."

With this: "by modeling these compartments as a one-dimensional segment 200 microns in length."

(iii) Aren't there two SD compartments?

Change: "The dynamics of the SD compartment only differs"

To: "The dynamics of the SD compartments only differ"

(6) Page 15:

Change: "that allows tau migrate"

To: "that allows tau to migrate"

Reviewer #2: In their manuscript, entitled “Emergence of directional bias in tau deposition from axonal transport dynamics” by Torok et al., the authors propose a first model to explain the alterations in axonal transport due to pathological Tau forms based on the differential impact of soluble and aggregated species. The model developed foresees the profile of tau distribution between a pair of pre- and post-synaptic neurons, and seems to be robust and stable through different initial conditions. Its elegancy lies on the fact that the ratio between a few of its parameters easily describes the biases recovered from the simulations. The authors present also an initial correlation to in vivo studies, and they also recognized the precariousness of this comparisons based on model´s limitations.

Altogether, the present work offers a good first model for understanding the initial states of AD and Non-AD related pathologies where the first affected neuron (with the axon compartment) passes on the pathological tau forms to the next one (or eventually previous one). This could serve as a starting point for designing future studies in order to gain molecular insight for tauopathies. The authors describe several restrictions of the model, such as a constant concentration of pathological tau species (n + m) over a period of days-months-years. Applicability of the models may increase if these corrections are introduced, as well as with the introduction of alternative boundary conditions. It would be of particular interest to analyze if the profiles are also reproduced at a larger scale, for example, introducing periodic or non-homogeneous conditions to simulate a larger “wire” of neurons with a mismatch between entry and exit of tau species at the SD compartments. However, the complexity of these models may require a dedicated manuscript.

A few minor concerns are described below:

- Section 2.2.2. Reconsider position of second parenthesis in the phrase: “…opposite direction (epsilon >> delta; Fig. 2b) and Vid. S2.”

- Labeling of supplementary videos SVideo_nob_regime.mp4 and SVideo_ret_regime.mp4 is inverted.

- Section 2.2.3. Second parenthesis is missing in the phrase “…both delta and epsilon values to be significantly greater than 0 (Fig. 2c and Vid.”

- Section 3.3. Referring to a plasma membrane as simple phospholipid bilayers is usually a big oversimplification. Considering the phrase “…migration across the two phospholipid bilayers of the SC…”, pre- and post-synaptic membranes are particularly of high complexity and its content of proteins is above average and not easy to overlook, particularly when referring to more complex mechanisms of transport for pathological Tau forms.

- Section 4.1.1. The expression “… and therefore demonstrate how the it influences model behavior…”, should be changed for “… and therefore demonstrate how it influences model behavior…”.

Reviewer #3: The authors are interested in investigating how different tau conformers may lead impact active transport and lead to different neurodegenerative pattern. They develop a 1-dimensional PDE model of tau protein transport in different axonal and synaptic compartments. Specifically, they simulate soluble and insoluble tau spreading between two neurons. The model has several strong simplifying assumptions that could be better justified. Model parameters (such as tau transport feedback, aggregation, fragmentation parameters) are varied to study the balance between anterograde and retrograde bias in the tau protein distribution. The authors apply this model to Alzheimer disease and non-AD in vivo mouse tauopathy models; the comparison is really to a network model from the same group, which should be made clearer. I appreciate that the authors provide a well-documented Github repository for their computational model.

Suggested revisions:

1. In most modeling studies for intracellular transport, effective transport is a quantity that emerges from the interplay of dynamics between anterograde and retrograde transport, diffusion, and pausing behavior. Here, it is used to mean an expression for velocity in the active transport state only as a function of two parameters introduced in this study. I would suggest reconsidering or clarifying the terminology to better connect to other studies.

2. As the authors mention in 4.1, the dynamics of motor proteins follows a start-and-stop dynamics; such a model has been validated with experimental data for axonal transport of various protein cargoes. It would be useful to mention in the paper how this simplifying assumption could affect the insights from this model.

3. I understand the need to keep the model 1-dimensional for computational efficiency. I recommend rephrasing some parts of section 4.1.2, such as “While the dynamics within axons can reasonably be collapsed into one dimension, those within the SD compartments cannot.” Since the authors make such a simplification to one dimension in this study (given the quantities of interest to them), they should instead start by explaining why their goals allow them to make this approximation.

4. In section 2.3, it would be useful to provide the bias measure in equation form. I am also unsure if figure 3b) best captures the insights from this section, but perhaps this is simply my stylistic preference.

5. For section 2.4: it is not surprising that a combination of the transport parameters influences the final state of the system, given the expression in equation (1).

6. In Table S1, it would help to add the literature sources for the model parameters used.

7. The authors repeatedly refer to “biases at the network level” in this manuscript. It seems that this is related to another preprint describing a network-level model from their group, but it is unclear what this phrasing means without reading that study. I think it is important that they clarify this and that they mention that the results are compared with results another modeling study (applied to experimental datasets). Since they compare directional biases across these models in Figure 6, I recommend that they also summarize how bias is quantified in that model.

8. In section 2.6, how were the parameters chosen in the simulations that match the directional bias data from in vivo studies? Did the authors carry out parameter estimation, or vary their parameters over some ranges? More details should be given here, especially since multiple parameter combinations might give good fits to this data.

**Have the authors made all data and (if applicable) computational code underlying the findings in their manuscript fully available?**

Reviewer #1: Yes

Reviewer #2: Yes

Reviewer #3: Yes

PLOS authors have the option to publish the peer review history of their article (what does this mean?). If published, this will include your full peer review and any attached files.

Reviewer #1: No

Reviewer #2: **Yes: **Lisandro Jorge Falomir Lockhart

Reviewer #3: No
---

## [Decision Letter · Decision Letter 1]

7 Jul 2021

Dear Mr. Torok,

We are pleased to inform you that your manuscript 'Emergence of directional bias in tau deposition from axonal transport dynamics' has been provisionally accepted for publication in PLOS Computational Biology.

Best regards,

Michele Migliore

Associate Editor

PLOS Computational Biology

Kim Blackwell

Deputy Editor

PLOS Computational Biology

Reviewer's Responses to Questions

**Comments to the Authors:**

Reviewer #2: Torok and collaborators have considerably improved their manuscript entitled “Emergence of directional bias in tau deposition from axonal transport dynamics”. The model proposed to explain the alterations in axonal transport due to pathological Tau forms is now easier to understand and has been better related to the profile of tau distribution between pre- and post-synaptic compartments. Addition of the section 4.3 in Methods y highly valuable. No further concerns remain for this reviewer.

Reviewer #3: The authors have taken into account the suggestions and have made changes to the text, methods, and figures that greatly improve the readability of the manuscript. Two small points:

- In the caption of figure 1, you may want to remove "within" in the second line.

- The caption of figure 2 refers to "each of the previous simulation conditions"; it may be helpful to specifically mention that the conditions studied here correspond to the simulation settings introduced in Figure 1.

**Have the authors made all data and (if applicable) computational code underlying the findings in their manuscript fully available?**

Reviewer #2: Yes

Reviewer #3: Yes

PLOS authors have the option to publish the peer review history of their article (what does this mean?). If published, this will include your full peer review and any attached files.

Reviewer #2: **Yes: **Lisandro Jorge Falomir Lockhart

Reviewer #3: No

---

## [Editor Report · Acceptance letter]

21 Jul 2021

PCOMPBIOL-D-21-00521R1 

Emergence of directional bias in tau deposition from axonal transport dynamics

Dear Dr Torok,

I am pleased to inform you that your manuscript has been formally accepted for publication in PLOS Computational Biology. Your manuscript is now with our production department and you will be notified of the publication date in due course.

With kind regards,

Andrea Szabo
